# Cervical Spine Pain and the Impact on the Quality of Life of Patients with Multiple Sclerosis

**DOI:** 10.3390/medicina60121923

**Published:** 2024-11-22

**Authors:** Martyna Odzimek, Hubert Lipiński, Małgorzata Błaszczyk, Patrycja Strózik, Julia Zegarek, Piotr Dubiński, Agata Michalska, Justyna Klusek, Marek Żak, Waldemar Brola

**Affiliations:** 1Doctoral School, The Jan Kochanowski University, Żeromskiego 5, 25-369 Kielce, Poland; odzimek.martyna@onet.pl (M.O.); piotr.dubinski@gmail.com (P.D.); 2Institute of Health Sciences, Collegium Medicum, The Jan Kochanowski University, al. IX Wieków Kielc 19A, 25-516 Kielce, Poland; hubertlipinski98@gmail.com (H.L.); gosiabl941@gmail.com (M.B.); patrycjastrozik321@gmail.com (P.S.); julia_zegarek@o2.pl (J.Z.); michalska.agata@ujk.edu.pl (A.M.); jsklusek@ujk.edu.pl (J.K.); mzak1@onet.eu (M.Ż.)

**Keywords:** multiple sclerosis (MS), cervical pain, quality of life (QoL)

## Abstract

*Background and Objectives*: The main aim of this study was to evaluate the impact of cervical pain on the quality of life of patients with multiple sclerosis in comparison with a group of healthy people (without diseases of the Central Nervous System). *Materials and Methods*: Data were collected at the Specialist Hospital St. Łukasz in Końskie (Poland) in the period from November 2023 to August 2024. The inclusion criteria for this study were as follows: age (20–50 years), women and men, healthy people (without diseases of the Central Nervous System) and people suffering from multiple sclerosis. People from the study group were diagnosed according to the McDonald criteria and tested with the EDSS. The mobility of the cervical spine was measured, and neck pain was assessed using the following: Visual Analogue Scale (VAS), Laitinen Scale and the Neck Disability Index (NDI). All participants self-assessed their quality of life using EuroQol 5D-5L (EQ-5D-5L). *Results*: 80 people took part in this study, the vast majority of whom were women (71.3%). The most common form of multiple sclerosis was relapsing-remitting (75.0%), and the average EDSS score was higher in the male group (1.6 ± 1.8). Cervical spine pain was reported by 27 people from the study group (67.5%) and 16 people from the control group (40.0%). In both groups, the cervical spine mobility was lower in people with neck pain. The level of cervical spine pain was statistically significantly (*p* < 0.05) higher in women, people living in small towns and in people with multiple sclerosis, depending on the type of disease and its duration. Cervical spine pain in people with multiple sclerosis was higher in all three scales (VAS mean = 5.7, ES = 0.79; Laitinen Scale mean = 10.1, ES = 0.60; and NDI Scale mean = 21.1, ES = 0.89). The study group obtained significantly higher scores on the EQ-5D scale (mean = 15.3; ES = 0.79) and EQ-VAS (mean = 53.2; ES = 0.94). *Conclusions*: This study proved that cervical spine pain is more common among people with multiple sclerosis. In these people, this problem is rarely diagnosed and properly treated.

## 1. Introduction

Pain in the cervical spine is one of the most common problems in the musculoskeletal system in adults and may affect from 16 to 75% of adults around the world [1]. Current scientific articles show that the lowest incidence rates occurred in Latin America, while the highest occurred in the regions of East Asia [2]. Problems in the cervical spine were more common in women than men, and the occurrence of pain increased proportionally with age [1]. This disease is characterized by a complex etiology. Several scientific studies indicate that the most common risk factors for cervical spine pain include older age, female gender, low social status and a history of spine pain (of any section) [2]. In addition, the factors that significantly influence cervical spine pain include psychological factors (anxiety, stress, fear, depression), cognitive factors (life attitudes, general beliefs, personality), sleep problems, social and professional support, style life (risky behaviours, the use of electronic devices, incorrect eating habits, the lack of physical activity) or genetic and disease factors (musculoskeletal diseases, autoimmune diseases, genetic diseases) [1,2,3,4,5]. Cervical spine pain is one of the most common causes of pain and disability in the everyday life of working people. It significantly affects physical, mental and social health, which may deteriorate and prevent patients from professional activity. Appropriate education, not only of sick people, but of the entire society, is important in the prevention of cervical spine pain [3,5]. Multiple sclerosis is a chronic, incurable disease of the Central Nervous System characterized by inflammation, demyelination, neuronal loss and gliosis [6]. It is estimated that 2.9 million people currently suffer from multiple sclerosis, of which 51,000 live in Poland [7]. Research results show that the disease more often affects women, and the average age of onset, according to various sources, ranges from 20 to 40 years of age [7,8]. The etiology of the disease is not known, but numerous scientific publications suggest that both environmental and genetic factors may influence the occurrence of the disease [6]. Depending on the course of the disease, we can distinguish the following forms: relapsing-remitting, primary-progressive, secondary-progressive and progressive-relapsing. Depending on the type, the disease may have different symptoms [9]. People suffering from multiple sclerosis most often struggle with the following symptoms: motor (paresis, spasticity), sensory (paresthesia, sensory disturbances: pain, vibration, touch, temperature), cerebellar (ataxia, tremor, nystagmus), autonomic (urination and/or bowel disorders, sexual disorders), mental (depression, cognitive disorders), brain stem (double vision, damage to the V and/or VII nerve) and others, such as fatigue [8,9]. The patient’s symptoms may vary depending on the stage of the disease, the location of the lesions, the duration of the disease and the treatment used. An interdisciplinary approach is important in the treatment of people with multiple sclerosis, allowing the patient to maintain functional fitness for as long as possible [8,9,10]. The main aim of this study was to evaluate the impact of cervical pain on the quality of life of patients with multiple sclerosis in comparison with a group of healthy people (without diseases of the Central Nervous System). The main hypothesis in our study is that cervical spine pain occurs more often in people with multiple sclerosis. This problem may significantly affect the quality of life of patients.

## 2. Materials and Methods

### 2.1. Ethics Statement and Information About Project

This study received positive consent from the Bioethics Committee of the Collegium Medicum of the Jan Kochanowski University (approval date: 30 June 2023, no. 32/2023). Before joining this study, each participant was informed about the objectives, methods, benefits and risks of participating in this study. All participants (both study and control groups) gave informed, voluntary consent to participate in this study, which was documented on consent forms available for review from the first author. Each study participant was covered by accident insurance (certificate no. COR422153). All procedures were performed in accordance with the 1964 Helsinki Declaration, as amended. This project was not financed by external institutions.

### 2.2. Study Population

A total of 80 people took part in this study based on random sampling, of which 40 people (50.0%) constituted the control group (healthy people without disorders of the Central Nervous System), while 40 people (50.0%) constituted the study group (patients with multiple sclerosis). This study was conducted at the Specialist Hospital of St. Łukasz in Końskie (Poland) from November 2023 to August 2024. The inclusion criteria for potential study participants were age (20–50 years), women and men, healthy people (without diagnosed disease syndromes or congenital defects of the Central Nervous System) and people with multiple sclerosis (with a confirmed diagnosis). The exclusion criteria were age (under 20 years of age or over 50 years of age), traffic accident, cervical spine and head injuries, head and/or neck surgery, the lack of consent to participate in this study, withdrawal from this study, the deterioration of health conditions of the research participant and unexpected random situations.

### 2.3. Methods

The research was conducted by a team consisting of neurologists and physiotherapists. All test results were performed during one visit by people trained in cervical spine diagnostics. The course of this study was standardized. The people participating in this study were examined on a fasting status, between 8:00 and 12:00 a.m., before work (if they worked) or before any physical exercise, performed in a sitting position. All the above-mentioned recommendations were made to reduce errors in pain perception, which allowed us to increase the transparency and repeatability of this study. The patients qualified for this study in the first stage completed an original questionnaire regarding socio-economic data. The questionnaire included questions about age, gender, the place of residence and a screening question regarding current neck pain. In addition, the range of motion of the cervical spine was measured in each person. The measurements of the range of motion of the cervical spine were obtained using a Baseline inclinometer (product code: 4372-4405E, measurement error ±0.5°) [11].

The criterion for including people from the study group in this study was a confirmed diagnosis of multiple sclerosis. The diagnosis was provided by the hospital, and the examination was performed by a neurologist. The diagnosis was based on the 2017 McDonald criteria [12]. The study group included patients with the following: relapsing-remitting multiple sclerosis (RRMS); primary progressive multiple sclerosis (PPMS); and secondary progressive multiple sclerosis (SPMS). All patients from the study group underwent a neurological examination and assessment on the Expanded Disability Status Scale (EDSS) [13]. This scale assesses the following: functions of the pyramidal system, functions of the cerebellum, functions of the brainstem, functions of the sensory system, functions of the intestines and urinary system, visual functions, mental functions, mobility abilities and other irregularities in functioning noted by a doctor. In the next part of this study, the responses of people participating in this study were analyzed in order to select people with current cervical spine pain. All people who marked ‘YES’ in the original questionnaire were subjected to further examinations of the cervical spine. People who marked ‘NO’ were not examined for cervical spine pain (only the range of motion of the cervical spine was measured) and only completed a quality of life survey. The individual scales and stages of this study are discussed below.

The first part of the cervical spine examination was based on a subjective assessment of pain. The Visual Analogue Scale (VAS) was used to subjectively assess the pain intensity. The interpretation is as follows: a value of 1–3 points for mild pain, 4–7 points for moderate pain, 8–9 points for severe pain and 10 points meant unbearable pain [14].

In the second part, patients assessed cervical spine pain according to the Laitinen Scale. This scale focuses on four aspects of pain rated from 0 points (no problem) to 4 points (maximum problem). Questions asked of the patient concern the following: pain intensity, the frequency of pain, the frequency of taking painkillers and limitations of motor activity. The results obtained were interpreted as follows: 0–4 points (0–25%)—no pain or minimal discomfort, 5–8 points (26–50%)—slight pain, which may affect the use of painkillers and limit activity motor skills, 9–12 points (51–75%)—moderate pain, which significantly affects the taking of painkillers and limitations of physical activity and 13–16 points (76–100%)—severe or unbearable, which occurs very often and causes the constant use of painkillers, making normal work and functioning in life impossible [15].

The last stage of the cervical spine examination was the assessment of pain using the Neck Disability Index (NDI). The main purpose of this tool was to assess the participants’ current cervical spine pain. It consists of 10 questions about everyday activities, scored from 0 points (no pain while performing the task) to 5 points (the worst possible pain you can imagine while performing the task). The scale includes questions about pain intensity, independence, lifting objects, activities of daily living, reading, headaches, concentration, work, driving, sleeping and recreation. The obtained results can be interpreted as follows: 0–4 points (0–8%)—no disability, 5–14 points (10–28%)—mild disability, 15–24 points (30–48%)—disability to a moderate degree, 25–34 points (50–64%)—a significant degree of disability and 35–50 points (70–100%)—a total degree of disability [16].

This study used devices and methods with proven scientific significance in detecting problems in the cervical spine. To properly interpret the obtained test results, three different scales with different sensitivities and specificities were used. This action allowed us to reliably look at the patient’s problem and check what factors influence the occurrence of pain.

The last part of this study was conducted based on the EQ-5D-5L questionnaire, which takes into account five dimensions of the quality of life. The following are assessed: the ability to move, self-service, the performance of everyday activities, feeling pain/discomfort; and feeling anxious/depressed. The scale ranges from 1 point (no problem) to 5 points (impossibility to perform the activity). The second part of the assessment was based on a Visual Scale Analogue (EQ-VAS, EuroQol Visual Analogue Scale), with which patients self-reported health conditions. This scale shows how good or bad of a day respondents had at the time of the survey. The scale ranges from 0 to 100, where 0—the worst imaginable health condition, and 100—the best imaginable state of health [17,18].

### 2.4. Statistical Analysis

Statistical analyzes were performed using Statistica™ version 13.3 (TIBCO Software Inc., Palo Alto, CA, USA) and advanced techniques in Microsoft Excel. G*power software version 3.1.9.7 (Düsseldorf, Germany) was used to calculate the statistical power of the sample size, and the total number of participants was calculated to be 72 (effect size = 0.9, error probability α = 0.05, power = 0.9). A total of 88 people took part in this study, of which 8 results were rejected due to formal deficiencies (the lack of consent, the lack of signature, incomplete documentation, an incorrectly completed questionnaire). Of the 80 people participating in this study, 40 people (50.0%) constituted the control group (healthy people without the Central Nervous System diseases), while 40 people (50.0%) constituted the study group (suffering from multiple sclerosis). Statistical description techniques and the Shapiro–Wilk normality test were used to describe groups and variables. The parametric Student’s *t* test was used for statistical analyses when the variables were normally distributed. If the variables had a non-normal distribution, the following statistical tests were used: U Mann–Whitney test, Wilcoxon test or Kruskall–Wallis test. In order to verify the hypotheses, quantitative variables (e.g., age) were used to compare groups, and, to compare groups in terms of nominal and ordinal variables, a contingency table analysis with a chi-square analysis was performed. This study assessed the magnitude and clinical significance of the results based on effect size statistics. According to Cohen’s indices, a value from 0 to 0.20 indicates a negligible effect size, a value from 0.21 to 0.5 indicates a small effect size, a value from 0.51 to 0.80 indicates a medium effect size and a value >0.80 indicates a large effect size. The *p* values ≤ 0.05 were considered statistically significant.

## 3. Results

In this study, 80 people participated, of which 40 people (50.0%) constituted the control group, and 40 people constituted the study group (50.0%). Women constituted the vast majority of this study (57 people, 71.3%), of which 31 women were in the study group (77.5%) and 26 women (65.0%) in the control group. Table 1 shows The group of men was smaller with 23 people (28.7%), including 9 people in the study group (22.5%), and 14 people in the control group (35.0%). In the study group, the majority of people were from the age group of 20–30 years (21 people, 52.5%), while, in the control group, the age groups 20–30 and 31–40 were equal (15 people, 37.5%). In the study group, the most common place of residence was a village (17 people, 42.5%), while, in the control group, a small town (24 people, 60.0%). There were no statistically significant differences in gender, age and place of residence (*p* > 0.05). A screening question about cervical spine pain showed that, in the group of people with multiple sclerosis, 27 people (67.5%, including 21 women and 6 men) and 16 people in the control group (40.0%, including 13 women and 3 men) reported the problem. The examined relationship is statistically significant (*p* < 0.05).

In the study group, most people suffered from relapsing-remitting (RRMS) multiple sclerosis (30 people, 75.0%), including 24 people (77.4%) from the group of women, and 6 people from the group of men (6 people, 66.7%). The fewest people were recorded in the secondary progressive (SPMS) multiple sclerosis—three people (7.5%), two women (6.5%) and one man (11.1%). The average EDSS index was higher in the men’s group (1.6 ± 1.8), while the disease duration was longer in the women’s group (5.5 years ± 3.2). There were no statistically significant differences depending on the type of disease, the EDSS index or disease duration (*p* > 0.05) (Table 2).

Table 3 presents detailed results of the cervical spine examination of all patients from the study and control groups. The mobility of the cervical spine was significantly lower in the study group than in the control group in all cervical spine movements (in both painless and painful patients). People with cervical spine pain in the study group had smaller ranges of motion than other people in this group. In the control group, no differences between groups were found. The mobility of the cervical spine in the control group was greater in people without neck pain, except for rotations to the right.

Table 4 presents a detailed comparison of the occurrence of pain intensity (according to the VAS cale) depending on the demographic data and clinical characteristics of the patients. In the study group, most women rated their current pain at 4–7 points (25.0%), while men rated it at 1–3 points (7.5%). In the control group, women rated the pain at 1–3 points (20.0%), while, among men, the differences were negligible. In the study group, most people reported pain at the age of 20–30 (35.0%), while, in the control group, at the age of 41–50 (20.0%). In both the study and control groups, pain was most often reported by rural residents (35.0% and 20.0%, respectively). A total of 60.0% of patients with the relapsing-remitting form reported neck pain, 85.6% of patients with the primary progressive form and 100.0% of patients with the secondary progressive form. It was the last group that reported the highest pain levels, ranging from 4 to 10 points on the VAS. People with disease duration from 0 to 5 years reported pain at the level of 1–3 points (28.6%) and 4–7 points (33.3%). In both the 6–10 year disease duration group and the 10+ year disease duration group, varying levels of neck pain were reported, which were higher than in those with a shorter disease duration. There was a statistically significant relationship between pain and the following: gender, living in a small town, the type of multiple sclerosis and disease duration (*p* < 0.05). There were no statistically significant differences in men, within age groups or depending on the EDSS score (*p* > 0.05).

Table 5 shows the pain level results for people who reported this problem during screening. These were 27 (67.5%) people from the study group and 16 (40.0%) people from the control group. The analysis was divided into groups by gender. The average cervical spine pain intensity score according to the VAS was higher in the group of people with multiple sclerosis (mean = 5.7) compared to the group of healthy people (mean = 3.5). Both in the study and control groups, women indicated a higher level of pain (6.3 and 3.8, respectively). Similar results were obtained according to the Laitinen scale. Higher results were reported in the group of multiple sclerosis patients (mean = 10.1) than in the control group (mean = 7.2). Higher results were reported in women, both in the study group (mean = 10.7) and in the control group (mean = 7.9). Higher rates of cervical disability were recorded in the study group (mean = 21.1) than in the control group (17.3). Women had higher scores on the NDI Scale, both in the study group (mean = 23.0) and in the control group (20.9).

In the last part of this study, the current quality of life of patients with multiple sclerosis and people from the control group was assessed. The average EQ-5D-5L score was higher in the group of people with multiple sclerosis (mean = 15.3) compared to the group of healthy people (mean = 8.4). Higher results were recorded in the pain group (mean = 19.1) with multiple sclerosis, while, in the control group, there was no significant difference between the groups (Table 6). The mean EQ-VAS score was higher in the control group (mean = 77.6) than in the study group (mean = 53.2). The lower results were recorded in the group of MS with pain.

## 4. Discussion

Pain in the cervical spine is one of the most frequently reported problems of the musculoskeletal system in adults [2,19]. It is estimated that this problem affects as many as 203 million people around the world to varying degrees [20]. Researcher forecasts indicate that, by 2050, this problem may affect up to 269 million people around the world, and the main cause of this phenomenon is the ageing of society [2,19,20]. However, the results of other studies indicate that the following factors may have a significant impact on the occurrence of cervical spine pain: stress [2,5,21], the lack of social support [1,2,5,19], anxiety [2,15], depression [2,5,21,22,23], comorbidities [1,2,22], low economic status [1], low levels of education [1] and work in an unergonomic position [1,5,24]. The results of scientific research show that women struggle with this problem more often [5,19,20], but this result evens out between the ages of 45 and 74 [20]. Other publications reduce the range of occurrence and state that they most often occur in the age groups 45–49 and 50–54, to the same extent in women and men [25]. There is no information in the scientific literature on the influence of the place of residence on the occurrence of neck pain. When it comes to the presence of pain in multiple sclerosis patients, research studies and clinical reports often show an inconsistent picture. Very often, the pain is neuropathic in nature and may affect up to 86% of patients in the form of the following: limb pain, neuralgia, spine pain or headaches [26]. A review of the literature largely indicates that patients have problems with back pain, and the incidence ranges from 8.6 to even 50% [27,28]. However, a serious problem is the lack of localization of the pain area [28]. Research conducted in Lithuania concerned the characteristics of pain and its relationship with the quality of life of patients with multiple sclerosis. The control group and the study group each consisted of 120 people (71.7% women, 28.3% men), and the average age of people with multiple sclerosis was 44.0 years. Most patients were diagnosed with relapsing-remitting disease, and men in the study group had a higher mean EDSS score (3.5 vs. 2.5). The average disease duration was 9.0 years. The authors paid particular attention to the fact that any type of pain occurred in as many as 92 people (76.7%) from the study group. People from the study and control groups rated the pain level at 4.0 according to the NRS (Numeric Rating Scale). An important conclusion from the conducted research is that cervical spine pain occurred much more often in people from the control group (9.2%) than in the study group (1.7%), and this relationship was statistically significant (*p* < 0.05). Our own research shows similar conclusions. The study group consisted of 40 people suffering from multiple sclerosis (77.5% women, 22.5% men), and the average age of patients was 35.4 years. Similarly to the above studies, the vast majority of people had a relapsing-remitting form of the disease (75.0%), and men had a higher average EDSS score (1.6 vs. 1.3). A visible difference between the studies is the duration of the disease, which was, on average, 5.3 years (5.5 years longer in women than in men). Our research contradicts that of Veličkaitė et al., as patients with multiple sclerosis reported cervical spine pain more often (67.5% vs. 40.0%) [29]. The main aim of another important study was to assess the intensity and location of pain in patients with multiple sclerosis. A total of 115 people took part in this study, the average age was 30.4 years and the majority of the group were women. Apart from the knees and wrists, the most common location of pain was the cervical spine (41.7%). It is worth emphasizing that women more often reported pain in this part of the spine, and the examined relationship was statistically significant [30]. This theory is also confirmed by the authors’ research, in which women reported cervical spine pain more often in both the study group (52.5%) and the control group (32.5%), and the obtained results were statistically significant (*p* < 0.05). However, other studies show that neck pain (3.2%) is of minor importance among other pain conditions experienced by patients with multiple sclerosis [31]. A study conducted by Łabuz-Roszak et al. included a group of 144 patients, the vast majority of whom, similarly to our study, were women, and the average age was 41 years [32]. Studies have shown that 50% of patients with multiple sclerosis struggle with back pain, and the pain is most often moderate or severe (average value 5.6). The disadvantage of the study was the lack of localization of spine pain, which is why the results obtained by the authors are difficult to compare with our own research. However, the results determining the impact of each type of pain on the quality of life of patients with multiple sclerosis are important. We have not found more information in the scientific literature about the impact of cervical spine pain on the quality of life of patients with multiple sclerosis. It is worth adding that scientific research indicates that BMI can significantly differentiate the health status of people with multiple sclerosis and influence the functional capacity of the musculoskeletal system [33].

It is important that researchers pay special attention to the impact of various diseases on the quality of life of patients. This action can significantly improve the quality of medical services provided and the psychophysical health of patients [34,35]. Moreover, the decline in cognitive functions may have a significant impact on the condition and quality of life of MS patients [36]. As mentioned earlier, the quality of life in people with multiple sclerosis decreased with the impact of depression, mood and anxiety [37,38]. Numerous environmental factors and lifestyle can significantly influence the occurrence of pain and, consequently, affect the quality of life. Post-traumatic changes such as differences in the size of the vertebrae, muscle damage, inflammatory changes and the degeneration of intervertebral discs, car accidents (e.g., whiplash injuries) or injuries during contact sports may intensify this problem [20]. Performing everyday activities such as lifting heavy objects or non-ergonomic body positions during work and repeating the same movements over and over (e.g., only in one direction) may significantly increase pain. It is worth adding that an incorrect sleeping position, an incorrectly selected pillow or an incorrect driving position may worsen problems in this area [39,40]. Additionally, scientific research shows that biological (genetic) factors may contribute to differences in pain perception. Additionally, it is worth noting that comparing two people will not always provide the same results due to the influence of factors such as genetics, microbiomes and life experiences [20,41]. It is important to introduce appropriate pain prevention measures in workplaces in order to obtain the most ergonomic position while working (a properly adjusted desk and chair, changes in workstation, taking breaks for exercise) to alleviate existing pain symptoms and prevent them from getting worse [42,43,44]. Appropriate diagnosis and treatment are also important. Access to doctors and imaging tests can contribute to higher health care costs. However, the quick detection of pain may contribute to a reduced number of sick leaves being issued and, consequently, to a reduction in work incapacity [45]. It is worth noting that symptoms affecting the assessment of the quality of life of people with multiple sclerosis may be perceived differently by the patient and by a neurologist. Therefore, it is important to improve communication to understand patients’ needs and their impact on quality of life [46].

This research provided important information about the impact of cervical spine pain on the quality of life of patients with multiple sclerosis. The comparison was made with a group of people without diseases of the Central Nervous System. However, there are limitations that may affect the reception of this study. Firstly, there is an unequal study and control group, which may mislead readers despite their high statistical significance. It would be important to introduce larger group sizes and take into account other potential variables. Researchers did not take into account the influence of genetic, socioeconomic, psychosocial and environmental factors as those that may influence the occurrence of cervical spine pain. A significant drawback of this study is the narrow age range of people qualified for this study (20–50 years), which may not show the full essence of the problem. The unequal distribution of groups of multiple sclerosis patients may suggest that the problem concerns only one type of disease. This study did not take into account the type of treatment taken by multiple sclerosis patients, which could have influenced the results. Moreover, it was a small-scale, single-centre study. It would be important to include participants from other centres and collect data on neck pain and its characteristics. Additionally, this study can be improved by using more extended scales (SF-12, SF-36, WHOQOL-BREF or MSQOL-54), which can show the specific, multifactorial nature of the impact of the disease on the quality of life. Extending the analyses with qualitative or longitudinal studies may increase the reliability of the results. It may be useful when planning research by other researchers.

## 5. Conclusions

The research results presented by our team are the first cross-sectional studies in Poland showing the incidence of cervical spine pain in patients with multiple sclerosis. This study showed that cervical spine pain occurred much more frequently in the study group (67.5% vs. 40.0%), which was not always consistent with the available scientific literature. People from the study group presented higher results in the VAS, Laitinen Scale and NDI Sales. In the study and control groups, the mobility of the cervical spine was significantly lower in people with the neck pain. This research may contribute to increasing scientists’ interest in research on the cervical spine in people with multiple sclerosis, which in turn may contribute to improving people’s quality of life. We suggest the need for interdisciplinary research to evaluate multiple sclerosis patient groups at the primary care level. In the research process, it is important to take into account genetic, psychosocial, socio-economic and environmental variables. It may be important to include exercise, physiotherapy and neck pain prevention advice in patients’ treatment and testing. A more accurate assessment of the prevalence of neck pain in multiple sclerosis patients in different geographic regions would enable the development of new treatment strategies. It is important to implement prevention and health promotion, which will help spread knowledge about combating neck pain.

## Figures and Tables

**Table 1 medicina-60-01923-t001:** Demographic characteristics of patients.

Characteristics	Study Group (With MS) *n* = 40, (%)	Control Group (Without MS) *n* = 40, (%)	Total, *n* = 80, (%)	*p*-Value *
Gender, *n* (%)				
Female	31 (77.5)	26 (65.0)	57 (71.3)	0.21
Male	9 (22.5)	14 (35.0)	23 (28.7)	
Age (years)				
20–30	21 (52.5)	15 (37.5)	36 (45.0)	0.35
31–40	10 (25.0)	15 (37.5)	25 (31.3)	
41–50	9 (22.5)	10 (25.0)	19 (23.7)	
Place of residence *n* (%)				
Small city	15 (37.5)	24 (60.0)	39 (48.8)	0.10
Big city	8 (20.0)	7 (17.5)	15 (18.8)	
Village	17 (42.5)	9 (22.5)	26 (32.4)	
Pain in the cervical spine				
Yes	27 (67.5)	16 (40.0)	43 (53.8)	**<0.05**
No	13 (32.5)	24 (60.0)	37 46.2)	

Note: MS—multiple sclerosis; statistically significant differences in bold; * statistical test selected from Section 2.4.

**Table 2 medicina-60-01923-t002:** Clinical characteristics of the study group.

Characteristics of Study Group	Female, *n* = 31, (%)	Male *n* = 9, (%)	Total, *n* = 40, (%)	*p*-Value *
Type of MS				
RRMS	24 (77.4)	6 (66.7)	30 (75.0)	0.79
PPMS	5 (16.1)	2 (22.2)	7 (17.5)	
SPMS	2 (6.5)	1 (11.1)	3 (7.5)	
EDSS score (mean ± SD)	1.3 (1.7)	1.6 (1.8)	1.3 (1.6)	0.31
Duration of disease (years, mean ± SD)	5.5 (3.2)	5.0 (2.5)	5.3 (3.1)	0.20

Note: MS—multiple sclerosis; EDSS—expanded disability status scale; SD—standard deviation; * statistical test selected from Section 2.4.

**Table 3 medicina-60-01923-t003:** Mobility of the cervical spine.

Measurements of Mobility of the Cervical Spine (Mean ± SD)	Study Group *n* = 40				Control Group *n* = 40			
	with Pain *n* = 27	Without Pain *n* = 13	Sum, *n* = 40	ES Cohen’s d	with Pain *n* = 16	Without Pain *n* =24	Sum, *n* = 40	ES Cohen’s d
Flexion	51.3 (12.7)	55.7 (11.8)	53.5 (12.3)	0.35	54.9 (10.1)	57.2 (10.5)	56.1 (10.3)	0.22
Extension	43.8 (10.7)	48.9 (9.3)	46.4 (10.0)	0.51	47.0 (13.0)	49.4 (11.3)	48.2 (12.2)	0.19
Rotation to the right	60.7 (14.8)	63.1 (12.9)	61.9 (13.9)	0.17	62.7 (13.9)	65.3 (12.1)	64.0 (13.0)	0.20
Rotation to the left	60.9 (13.0)	61.5 (13.4)	61.2 (13.2)	0.14	62.3 (13.7)	62.0 (13.2)	62.2 (13.5)	0.02
Lateral flexion to the right	39.1 (8.6)	42.5 (9.0)	40.8 (8.8)	0.38	40.3 (7.7)	42.2 (6.9)	41.3 (7.3)	0.25
Lateral flexion to the left	39.3 (9.0)	41.1 (9.2)	40.2 (9.1)	0.19	39.9 (8.7)	41.0 (8.2)	40.5 (8.5)	0.13

Note: SD—standard deviation; ES—effect size.

**Table 4 medicina-60-01923-t004:** The cervical spine pain intensity (VAS) depending on patient demographic and clinical characteristics.

Characteristics	Study Group (with MS) *n* = 40, (%) VAS (0–10 Points)					Control Group (Without MS) *n* = 40, (%) VAS (0–10 Points)					*p*-Value *
	0	1–3	4–7	8–9	10	0	1–3	4–7	8–9	10	
Gender, *n* (%)											
Female	10 (25.0)	5 (12.5)	10 (25.0)	4 (10.0)	2 (5.0)	13 (32.5)	8 (20.0)	2 (5.0)	2 (5.0)	1 (2.5)	**<0.05**
Male	3 (7.5)	3 (7.5)	1 (2.5)	1 (2.5)	1 (2.5)	11 (27.5)	1 (2.5)	2 (5.0)	0 (0.0)	0 (0.0)	0.49
Age (years)											
20–30	7 (17.5)	5 (12.5)	6 (15.0)	3 (7.5)	0 (0.0)	12 (30.0)	2 (5.0)	1 (2.5)	0 (0.0)	0 (0.0)	0.87
31–40	4 (10.0)	2 (5.0)	2 (5.0)	1 (2.5)	1 (2.5)	10 (25.0)	3 (7.5)	2 (5.0)	0 (0.0)	0 (0.0)	0.81
41–50	2 (5.0)	1 (2.5)	3 (7.5)	1 (2.5)	2 (5.0)	2 (5.0)	4 (10.0)	1 (2.5)	2 (5.0)	1 (2.5)	0.83
Place of residence, *n* (%)											
Small city	5 (12.5)	4 (10.0)	3 (7.5)	2 (5.0)	1 (2.5)	18 (45.0)	5 (12.5)	1 (2.5)	0 (0.0)	0 (0.0)	**<0.05**
Big city	5 (12.5)	1 (2.5)	1 (2.5)	1 (2.5)	0 (0.0)	5 (12.5)	1 (2.5)	1 (2.5)	0 (0.0)	0 (0.0)	0.57
Village	3 (7.5)	3 (7.5)	7 (17.5)	2 (5.0)	2 (5.0)	1 (2.5)	3 (7.5)	2 (5.0)	2 (5.0)	1 (2.5)	0.69
Type of MS											
RRMS	12 (30.0)	6 (15.0)	8 (20.0)	2 (5.0)	2 (5.0)	Not applicable.					**<0.05**
PPMS	1 (2.5)	2 (5.0)	2 (5.0)	2 (5.0)	0 (0.0)						
SPMS	0 (0.0)	0 (0.0)	1 (2.5)	1 (2.5)	1 (2.5)						
EDSS Score											
0–5	12 (30.0)	5 (12.5)	8 (20.0)	3 (7.5)	2 (5.0)	Not applicable.					0.48
6–10	1 (2.5)	3 (7.5)	3 (7.5)	2 (5.0)	1 (2.5)						
Duration of disease											
0–5 years	8 (20.0)	6 (15.0)	7 (17.5)	0 (0.0)	0 (0.0)	Not applicable.					**<0.05**
6–10 years	4 (10.0)	1 (2.5)	3 (7.5)	3 (7.5)	1 (2.5)						
More than 10 years	1 (2.5)	1 (2.5)	1 (2.5)	2 (5.0)	2 (5.0)						

Note: MS—multiple sclerosis; statistically significant differences in bold; * statistical test selected from Section 2.4.

**Table 5 medicina-60-01923-t005:** Results of cervical spine pain depending on gender.

Results of Cervical Spine Examination	Study Group (with MS) –with Pain			Control Group (Without MS) –with Pain			Total, *n* = 43	ES Cohen’s d
	Female *n* = 21	Male *n* = 6	Sum, *n* = 27	Female *n* = 13	Male *n* =3	Sum, *n* = 16		
VAS (mean ± SD)	6.3 (4.5)	3.5 (2.0)	5.7 (3.9)	3.8 (2.0)	2.9 (1.9)	3.5 (2.1)	4.9 (2.4)	0.79
Laitinen Scale (mean ± SD)	10.7 (2.6)	8.3 (1.7)	10.1 (2.6)	7.9 (2.5)	5.7 (3.0)	7.2 (2.9)	8.1 (1.8)	0.60
NDI Scale (mean ± SD)	23.0 (9.4)	14.4 (7.7)	21.1 (9.7)	20.9 (5.6)	13.6 (7.7)	10.8 (6.5)	17.3 (6.5)	0.89

Note: MS—multiple sclerosis; SD—standard deviation; ES—effect size.

**Table 6 medicina-60-01923-t006:** Results of the EQ-5D-5L questionnaire.

EQ-5D	Study Group (with MS)			Control Group (Without MS)			Total, *n* = 80	ES Cohen’s d
	with Pain *n* = 27	Without Pain *n* = 13	Sum, *n* = 40	with Pain *n* = 16	Without Pain *n* = 24	Sum, *n* = 40		
EQ-5D (mean ± SD)	19.1 (8.2)	10.3 (9.1)	15.3 (9.9)	8.7 (6.9)	7.9 (7.6)	8.4 (7.3)	11.5 (8.5)	0.79
EQ-VAS (mean ± SD)	51.6 (26.8)	58.3 (26.9)	53.2 (27.0)	77.1 (19.1)	78.6 (18.5)	77.6 (18.9)	66.4 (22.8)	0.94

Note: MS—multiple sclerosis; SD—standard deviation; ES—effect size.

## Data Availability

The data presented in this study are available from the first author upon request.

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
