# Peer review of "Cervical Spine Pain and the Impact on the Quality of Life of Patients with Multiple Sclerosis"

_medicina, 2024, doi:10.3390/medicina60121923_

Round 1
Reviewer 1 Report
Comments and Suggestions for Authors
The article is generally well done. Here are some suggestions for improvement:
It is suggested to review the formulation of some sentences to improve clarity. For example, in the abstract, the phrase "the impact on the quality of life compared to a group of healthy people" could be reformulated for greater precision, such as "to evaluate the impact of cervical pain on the quality of life of patients with multiple sclerosis in comparison with a group of healthy people."
Consider improving the introduction by providing a clearer structure regarding the relevance of the problem and the specific objectives of the study. This would facilitate understanding of the rationale behind the research.
It is recommended to add more details about the data collection process, such as whether patients were evaluated under controlled conditions (time of day, fasting status, etc.), to reduce biases in pain perception, thus enhancing the study's transparency and reproducibility.
The study could benefit from a more detailed description of the demographic data, discussing, for example, the relationship between pain levels and age within subgroups. Additionally, including an interpretation of the results from a clinical perspective would be beneficial to highlight the impact of the findings, such as the possible relationship between pain levels, treatment adherence, or patient independence.
It is suggested to delve into possible reasons for differences between your results and those of other studies, which would help contextualize the findings. You could also propose hypotheses on how environmental or lifestyle factors may influence the results, supported by additional literature.
The conclusion is clear, though it could be strengthened by including specific recommendations for future research or clinical implications. This would provide a clinically oriented conclusion.
If possible, consider adding more recent references and making more explicit connections between the existing literature and the results obtained in the study. This could increase the robustness of the justification and analysis.
The limitations section covers important aspects, but it would be valuable to add a reflection on potential biases arising from the sample selection process or sample size. Additionally, suggesting qualitative or longitudinal studies as complements to the current results could be valuable for future research.
Good job!
Author Response
REVIEW REPORT 1.
Dear Reviewer,
First of all, we would like to thank you for taking the time to read our article. We are very pleased that you liked the article. Due to the comments you sent us, we have tried to make corrections as best as possible. We hope that the changes introduced have helped us improve our work.
Point 1: It is suggested to review the formulation of some sentences to improve clarity. For example, in the abstract, the phrase “the impact on the quality of life compared to a group of healthy people” could be reformulated for greater precision, such as "to evaluate the impact of cervical pain on the quality of life of patients with multiple sclerosis in comparison with a group of healthy people."
Response 1: Thank you very much for your comment and we agree with the reviewer. All phrases regarding the main purpose of the work have been changed according to the instructions. We encourage you to familiarize yourself with the changes introduced.
Abstract: „Background: The main aim of the study was to evaluate the impact of cervical pain on the quality of life of patients with multiple sclerosis in comparison with a group of healthy people (without diseases of the Central Nervous System)”.
Introduction: ,,The main aim of the study was to evaluate the impact of cervical pain on the quality of life of patients with multiple sclerosis in comparison with a group of healthy people (without diseases of the Central Nervous System)”.
Discussion: ,, The research provided important information about the impact of cervical spine pain on the quality of life of patients with multiple sclerosis. The comparison was made with a group of people without diseases of the Central Nervous System”.
Point 2: Consider improving the introduction by providing a clearer structure regarding the relevance of the problem and the specific objectives of the study. This would facilitate understanding of the rationale behind the research.
Response 2: Thank you so much for comment. We agree with the Reviewer's opinion and changes have been introduced in the abstract.
„Abstract: Background: The main aim of the study was to evaluate the impact of cervical pain on the quality of life of patients with multiple sclerosis in comparison with a group of healthy people (without diseases of the Central Nervous System). Methods: Data were collected at the Specialist Hospital St. Łukasz in Końskie (Poland) in the period from November 2023 to August 2024. The inclusion criteria for the study were: age (20-50 years), women and men, healthy people (without diseases of the Central Nervous System) and people suffering from sclerosis scattered. People from the study group were diagnosed according to the McDonald criteria and tested with the EDSS scale. The mobility of the cervical spine was measured and neck pain was assessed using: visual analogue scale (VAS), Laitinen scale and the Neck Disability Index (NDI). All participants self-assessed their quality of life using EuroQol 5D-5L (EQ-5D -5L). Results: In the study took part in 80 people, the vast majority of which were women (71.3%). The most common form of multiple sclerosis was relapsing-remitting (75.0%), and the average EDSS score was higher in the male group (1.6 ± 1.8). The cervical spine pain was reported by 27 people from the study group (67.5%) and 16 people from the control group (40.0%). The mobility of the cervical spine was lower in the study group, and statistically significant differences were noted in the movements of flexion, extension and lateral bending to the right (p < 0.05, ES = 0.35 - 0.51). The level of cervical spine pain was statistically significantly (p < 0.05) higher in women, people living in small towns, and in people with multiple sclerosis, depending on the type of disease and its duration. The cervical spine pain in people with MS was higher on all three scales (p < 0.05, ES = 0.79 – 0.89). The study group obtained significantly higher results on the EQ-5D scale (p < 0.05, ES = 0.79) and EQ-VAS (p < 0.001, ES = 0.94). Conclusions: The study proved that cervical spine pain is more common among people with multiple sclerosis. In these people, this problem is rarely diagnosed and properly treated.
Keywords: multiple sclerosis (MS); cervical pain; quality of life (QoL)”.
Point 3: It is recommended to add more details about the data collection process, such as whether patients were evaluated under controlled conditions (time of day, fasting status, etc.), to reduce biases in pain perception, thus enhancing the study's transparency and reproducibility.
Response 3: We agree with the Reviewer's comment that the description of the study procedure contained too little detail and standardization of the study for each subject. We have therefore improved this aspect in Section 2.3 (Methods).
„The research was conducted by a team consisting of neurologists and physiotherapists. All test results were performed during one visit by people trained in cervical spine diagnostics. The course of the study was standardized. The people participating in the study were examined on fasting status, between 8:00-12:00 a.m., before work (if they worked) or before any physical exercise, performed in a sitting position. All the above-mentioned recommendations were made to reduce errors in pain perception, which allowed to increase the transparency and repeatability of the study. The patients qualified for the study in the first stage completed an original questionnaire regarding socio-economic data. The questionnaire included questions about age, gender, place of residence and a screening question regarding current neck pain. In addition, the range of motion of the cervical spine was measured in each person. The measurements of the range of motion of the cervical spine using a Baseline inclinometer (product code: 4372-4405E, measurement error ±0.5◦) [11]”.
Point 4: The study could benefit from a more detailed description of the demographic data, discussing, for example, the relationship between pain levels and age within subgroups. Additionally, including an interpretation of the results from a clinical perspective would be beneficial to highlight the impact of the findings, such as the possible relationship between pain levels, treatment adherence, or patient independence.
Response 4: Thank you very much for the Reviewer's suggestion and for drawing attention to the multifactorial aspect of cervical spine pain. Therefore, we supplemented the study with additional tables and statistical data.
,, Table 4 presents a detailed comparison of the occurrence of pain intensity (according to the VAS scale) depending on the demographic data and clinical characteristics of the patients. There was a statistically significant relationship between pain and: gender, living in a small town, type of multiple sclerosis and disease duration (p < 0.05). There were no statistically significant differences in men, within age groups or depending on the EDSS scale score (p > 0.05).
Table 4. The cervical spine pain intensity (VAS Scale) depending on patient demographic and clinical characteristics.
|
Characteristics |
Study group (with MS) n = 40, (%) VAS Scale (0-10 point) |
Control group (without MS) n = 40, (%) VAS Scale (0-10 point) |
p-Value* |
||||||||
|
0 |
1-3 |
4-7 |
8-9 |
10 |
0 |
1-3 |
4-7 |
8-9 |
10 |
||
|
Gender, n (%) |
|
|
|||||||||
|
Female |
10 (25.0) |
5 (12.5) |
10 (25.0) |
4 (10.0) |
2 (5.0) |
13 (32.5) |
8 (20.0) |
2 (5.0) |
2 (5.0) |
1 (2.5) |
<0.05 |
|
Male |
3 (7.5) |
3 (7.5) |
1 (2.5) |
1 (2.5) |
1 (2.5) |
11 (27.5) |
1 (2.5) |
2 (5.0) |
0 (0.0) |
0 (0.0) |
0.49 |
|
Age (years) |
|
|
|||||||||
|
20-30 |
7 (17.5) |
5 (12.5) |
6 (15.0) |
3 (7.5) |
0 (0.0) |
12 (30.0) |
2 (5.0) |
1 (2.5) |
0 (0.0) |
0 (0.0) |
0.87 |
|
31-40 |
4 (10.0) |
2 (5.0) |
2 (5.0) |
1 (2.5) |
1 (2.5) |
10 (25.0) |
3 (7.5) |
2 (5.0) |
0 (0.0) |
0 (0.0) |
0.81 |
|
41-50 |
2 (5.0) |
1 (2.5) |
3 (7.5) |
1 (2.5) |
2 (5.0) |
2 (5.0) |
4 (10.0) |
1 (2.5) |
2 (5.0) |
1 (2.5) |
0.83 |
|
Place of residence, |
|
|
|||||||||
|
Small city |
5 (12.5) |
4 (10.0) |
3 (7.5) |
2 (5.0) |
1 (2.5) |
18 (45.0) |
5 (12.5) |
1 (2.5) |
0 (0.0) |
0 (0.0) |
<0.05 |
|
Big city |
5 (12.5) |
1 (2.5) |
1 (2.5) |
1 (2.5) |
0 (0.0) |
5 (12.5) |
1 (2.5) |
1 (2.5) |
0 (0.0) |
0 (0.0) |
0.57 |
|
Village |
3 (7.5) |
3 (7.5) |
7 (17.5) |
2 (5.0) |
2 (5.0) |
1 (2.5) |
3 (7.5) |
2 (5.0) |
2 (5.0) |
1 (2.5) |
0.69 |
|
Type of MS |
|
|
|||||||||
|
RRMS |
12 (30.0) |
6 (15.0) |
8 (20.0) |
2 (5.0) |
2 (5.0) |
Not applicable. |
<0.05 |
||||
|
PPMS |
1 (2.5) |
2 (5.0) |
2 (5.0) |
2 (5.0) |
0 (0.0) |
||||||
|
SPMS |
0 (0.0) |
0 (0.0) |
1 (2.5) |
1 (2.5) |
1 (2.5) |
||||||
|
EDSS Score |
|
|
|||||||||
|
0-5 |
12 (30.0) |
5 (12.5) |
8 (20.0) |
3 (7.5) |
2 (5.0) |
Not applicable. |
0.48 |
||||
|
6-10 |
1 (2.5) |
3 (7.5) |
3 (7.5) |
2 (5.0) |
1 (2.5) |
||||||
|
Duration of disease |
|
|
|||||||||
|
0-5 years |
8 (20.0) |
6 (15.0) |
7 (17.5) |
0 (0.0) |
0 (0.0) |
Not applicable. |
<0.05 |
||||
|
6-10 years |
4 (10.0) |
1 (2.5) |
3 (7.5) |
3 (7.5) |
1 (2.5) |
||||||
|
More than 10 years |
1 (2.5) |
1 (2.5) |
1 (2.5) |
2 (5.0) |
2 (5.0) |
||||||
Note: MS – multiple sclerosis; NS - not significant; statistically significant differences in bold; *statistical test selected from section 2.4.”
Point 5: It is suggested to delve into possible reasons for differences between your results and those of other studies, which would help contextualize the findings. You could also propose hypotheses on how environmental or lifestyle factors may influence the results, supported by additional literature.
Response 5: We agree with the Reviewer's comment and have tried to improve the article with additional information regarding the impact of environmental factors on the study results. Thank you very much for this important comment. The changes were introduced in Chapter 4 (Discussion).
,, It is important that researchers pay special attention to the impact of various diseases on the quality of life of patients. This can significantly improve the quality of medical services provided and the psychophysical health of patients [34,35]. Moreover, the decline in cognitive functions may have a significant impact on the condition and quality of life of MS patients [36]. As mentioned earlier, the quality of life in people with multiple sclerosis decreased with the impact of depression, mood and anxiety [37,38]. Numerous environmental factors and lifestyle can significantly influence the occurrence of pain and, consequently, affect the quality of life. Post-traumatic changes such as differences in the size of the vertebrae, muscle damage, inflammatory changes and de-generation of intervertebral discs, car accidents (e.g. whiplash injuries) or injuries during contact sports may intensify this problem [39]. Performing everyday activities such as lifting heavy objects or non-ergonomic body position during work, repeating the same movements over and over (e.g. only in one direction) may significantly increase pain. It is worth adding that an incorrect sleeping position, an incorrectly selected pillow or an incorrect driving position may worsen problems in this area [40,41]. Additionally, scientific research shows that biological (genetic) factors may contribute to differences in pain perception. Additionally, it is worth noting that comparing two people will not always provide the same results due to the influence of factors such as genetics, microbiome, experiences and life experiences [39,42]. It is important to introduce appropriate pain prevention measures in workplaces in order to obtain the most ergonomic position while working (properly adjusted desk and chair, change of workstation, taking breaks for exercise) to alleviate existing pain symptoms and prevent them from getting worse [43,44 .45]. Appropriate diagnosis and treatment are also important. Access to doctors and imaging tests can contribute to higher health care costs. However, quick detection of pain may contribute to a reduced number of sick leaves being issued and, consequently, to a reduction in work incapacity [46]. It is worth noting that symptoms affecting the assessment of the quality of life of people with multiple sclerosis may be perceived differently by the patient and by a neurologist. Therefore, it is important to improve communication to understand patients' needs and their impact on quality of life [47]”.
Point 6: The conclusion is clear, though it could be strengthened by including specific recommendations for future research or clinical implications. This would provide a clinically oriented conclusion.
Response 6: Thank you for drawing attention to such an important aspect. We have completed the missing information.
„We suggest that interdisciplinary research is needed to evaluate groups of patients with multiple sclerosis at the primary care level. It is important to take into account genetic, psychosocial, socioeconomic and environmental variables in the research process. Incorporating exercise, physiotherapy and advice on preventing neck pain into treatment and testing may be important. A more accurate assessment of the prevalence of neck pain in MS patients in different geographic regions would enable the development of new treatment strategies. It is important to implement prevention and health promotion, which will help spread knowledge about combating neck pain.”.
Point 7: If possible, consider adding more recent references and making more explicit connections between the existing literature and the results obtained in the study. This could increase the robustness of the justification and analysis.
Response 7: Due to the Reviewer's comments, we added new scientific publications to the analysis and changed the structure of Chapter 4 (Discussion).
„[35] Culicetto, L.; Lo Buono, V.; Donato, S.; La Tona, A.; Cusumano, A.M.S.; Corello, G.M.; Sessa, E.; Rifici, C.; D’Aleo, G.; Quartarone, A.; Marino, S. Importance of Coping Strategies on Quality of Life in People with Multiple Sclerosis: A Systematic Review. J Clin Med. 2024, 13 (18): 5505. DOI: 10.3390/jcm13185505.
[36] Marafioti, G.; Cardile, D.; Culicetto, L.; Quartarone, A.; Lo Buono, V. The Impact of Social Cognition Deficits on Quality of Life in Multiple Sclerosis: A Scoping Review. Brain Sci. 2024, 14 (7): 691. DOI: 10.3390/brainsci14070691.
[37] Valentine T.R..; Kuzu, D.; Kratz, A.L. Coping as a Moderator of Associations Between Symptoms and Functional and Affective Outcomes in the Daily Lives of Individuals With Multiple Sclerosis. Ann. Behav. Med. 2023, 57: 249–259. DOI: 10.1093/abm/kaac050.
[38] Ouwerkerk, M.; Rietberg, M.B.; van der Linden, M.M.W.; Uitdehaag, B.M.J.; van Wegen, E.E.H.; de Groot, V. Int J MS Care. 2024, 26 (Q3): 199–206. DOI: 10.7224/1537-2073.2023-060.
[39] Wu, A-M.; Cross, M.; Elliot, J.M.; Culbreth, G.T.; Haile, L.M.; Steinmetz, J.D.; Hagins, H.; Kopec, J.A.; Brooks, P.M.; Woolf, A.D.; Kopansky-Goles, D.R. et. al. Global, regional, and national burden of neck pain, 1990–2020, and projections to 2050: a systematic analysis of the Global Burden of Disease Study 2021. Lancet Rheumatol. 2024, 6: e142–55. DOI: 10.1016/S2665-9913(23)00321-1.
[40] Elliott, J.M.; Smith, A.C.; Hoggarth, M.A.; Albin, S.A.; Weber 2nd, K.A.; Haager, M.; Fundaun, J.; Wasilewski, M.; Courtney, D.M.; Parrish, T.B. Muscle fat infiltration following whiplash: A computed tomography and magnetic resonance imaging comparison. PLoS One. 2020, 15 (6): e0234061. DOI: 10.1371/journal.pone.0234061.
[41] Fortin, M.; Dobrescu, O.; Courtemanche, M.; Sparrey, C.J.; Santaguida, C.; Fehlings, M.G.; Weber, M.H. Association Between Paraspinal Muscle Morphology, Clinical Symptoms, and Functional Status in Patients With Degenerative Cervical Myelopathy. Spine (Phila Pa 1976). 2017, 42 (4): 232-239. DOI: 10.1097/BRS.0000000000001704.
[42] Mittinty, M.M.; Lee, J.Y.; Walton, D.M.; El-Omar, E.M.; Elliot, J.M. Integrating the Gut Micro-biome and Stress-Diathesis to Explore Post-Trauma Recovery: An Updated Model. Pathogens. 2022, 11 (7): 716. DOI: 10.3390/pathogens11070716.
[43] Jun, D.; Zoe, M.; Johnston, V.; O’Leary, S. Physical risk factors for developing non-specific neck pain in office workers: a systematic review and meta-analysis. Int Arch Occup Environ Health. 2017, 90 (5): 373-410. DOI: 10.1007/s00420-017-1205-3.
[44] Johnston, V.; Chen, X.; Welch, A.; Sjøgaard, G.; Comans, T.A.; McStea, M.; Straker, L.; Melloh, M.; Pereira, M.; O’Leary, S. A cluster-randomized trial of workplace ergonomics and neck-specific exercise versus ergonomics and health promotion for office workers to manage neck pain - a secondary outcome analysis. BMC Musculoskelet Disord. 2021, 22 (1): 68. DOI: 10.1186/s12891-021-03945-y.
[45] de Zoete, R.M.; Armfield, N.R.; McAuley, J.H.; Chen, K.; Sterling, M. Comparative effectiveness of physical exercise interventions for chronic non-specific neck pain: a systematic review with network meta-analysis of 40 randomised controlled trials. Br J Sports Med. 2020, bjsports-2020-102664. DOI: 10.1136/bjsports-2020-102664.
[46] Costello, J.E.; Shah, L.M.; Peckham, M.E.; Hutchins, T.A.; Anzai, Y. Imaging Appropriateness for Neck Pain. J Am Coll Radiol. 2020, 17 (5): 584-589. DOI: 10.1016/j.jacr.2019.11.005.
[47] Ysrraelit, M.C.; Fiol, M.P.; Gaitán, M.I.; Correale J. Quality of Life Assessment in Multiple Sclerosis: Different Perception between Patients and Neurologists. Front Neurol. 2018, 8: 729. DOI: 10.3389/fneur.2017.00729.”.
Point 8: The limitations section covers important aspects, but it would be valuable to add a reflection on potential biases arising from the sample selection process or sample size. Additionally, suggesting qualitative or longitudinal studies as complements to the current results could be valuable for future research.
Response 8: Thank you very much for such an important comment with which we agree. We added reflection on potential biases resulting from the sample selection process or sample size. In addition, we have indicated a further path for the development of qualitative or longitudinal research.
„The research provided important information about the impact of cervical spine pain on the quality of life of patients with multiple sclerosis. The comparison was made with a group of people without diseases of the Central Nervous System. However, there are limitations that may affect the reception of the study. Firstly, there is an unequal study and control group, which may mislead readers despite their high statistical significance. It would be important to introduce larger group sizes and take into account other potential variables. Researchers did not take into account the influence of genetic, socioeconomic, psychosocial and environmental factors as those that may influence the occurrence of cervical spine pain. A significant drawback of the study is the narrow age range of people qualified for the study (20–50 years), which may not show the full essence of the problem. The unequal distribution of groups of multiple sclerosis patients may suggest that the problem concerns only one type of disease. The study did not take into account the type of treatment taken by multiple sclerosis patients, which could have influenced the results. Moreover, it was a small-scale, single-center study. It would be important to include participants from other centers and collect data on neck pain and its characteristics. Additionally, the study can be im-proved by using more extended scales (SF-12, SF-36, WHOQOL-BREF or MSQOL-54), which can show the specific, multifactorial nature of the impact of the disease on the quality of life. Extending the analyzes with qualitative or longitudinal studies may increase the reliability of the results. It may be useful when planning research by other researchers.”.
Reviewer 2 Report
Comments and Suggestions for Authors
Dear authors,
Your study has a major methodological flaw that needs to be addressed to make it suitable for publication. The flaw lies in the lack of clarity regarding what you intended to achieve with this study. You state that you aimed to investigate back pain in MS patients compared to healthy people. However, you need to define what you mean by "healthy people." Are these individuals without MS but with back pain, or are they people with no MS and no back pain? Your manuscript contains no information regarding this definition or the process by which these healthy individuals were recruited. This is a major source of bias in your study, as if your groups are not comparable, then the study conclusions are invalid.
To elaborate further, if your control group consisted of healthy individuals with no MS and no back pain, what is the novelty of your study? The fact that MS is associated with back pain and that both conditions, together or separately, affect quality of life is not novel and is widely known and recognized. On the other hand, if your healthy individuals were people without MS but with back pain, why were there so few of them? If this is the case, all comparisons between the study and control groups are meaningless, as the composition of the control group was inadequate. Please consider addressing this issue.
I also have some minor comments:
- You state that your continuous variables were normally distributed, and therefore you selected parametric tests, which resulted in significant/highly significant differences between your study groups. However, this seems like an attempt to manipulate the data. For most continuous variables, the reported standard deviation is too wide relative to the mean, which suggests a platykurtic data distribution, which by definition is non-normal. In general, with such small comparison groups and an unhealthy population, it is highly unlikely to observe normal data distribution. Even if the data distribution was normal in the control group, it was obviously non-normal in the study group, and to apply parametric tests, both groups should be normally distributed. Please reconsider this approach.
- Subsection 2.3, titled "Methods," needs to be divided into several parts as it is currently messy and difficult to follow. Additionally, the authors need to explain why they used a variety of scales that appear to duplicate each other. What was the rationale for this?
Author Response
REVIEW REPORT 2.
Dear Reviewer,
Thank you very much for taking the time to check our work. We have tried our best to improve our work and respond to your comments. We hope that we have resolved the disputes as best as possible and clarified all inaccuracies.
Point 1: Your study has a major methodological flaw that needs to be addressed to make it suitable for publication. The flaw lies in the lack of clarity regarding what you intended to achieve with this study. You state that you aimed to investigate back pain in MS patients compared to healthy people. However, you need to define what you mean by "healthy people." Are these individuals without MS but with back pain, or are they people with no MS and no back pain? Your manuscript contains no information regarding this definition or the process by which these healthy individuals were recruited. This is a major source of bias in your study, as if your groups are not comparable, then the study conclusions are invalid.
Response 1: Thank you very much for such an important and relevant comment. When referring to the term ‘healthy people’, we meant people without MS. The main criteria for inclusion in the control group were age (20-50 years) and the absence of diseases of the Central Nervous System (in this case, MS). Therefore, we have supplemented our description and made appropriate changes to the summary and chapter 2.3 (Methods), where we described the entire process of recruiting people to the study and the inclusion and exclusion criteria.
,, The research was conducted by a team consisting of neurologists and physiotherapists. All test results were performed during one visit by people trained in cervical spine diagnostics. The course of the study was standardized. The people participating in the study were examined on fasting status, between 8:00-12:00 a.m., before work (if they worked) or before any physical exercise, performed in a sitting position. All the above-mentioned recommendations were made to reduce errors in pain perception, which allowed to increase the transparency and repeatability of the study. The patients qualified for the study in the first stage completed an original questionnaire regarding socio-economic data. The questionnaire included questions about age, gender, place of residence and a screening question regarding current neck pain. In addition, the range of motion of the cervical spine was measured in each person. The measurements of the range of motion of the cervical spine using a Baseline inclinometer (product code: 4372-4405E, measurement error ±0.5◦) [11].
The criterion for including people from the study group in the study was a confirmed diagnosis of multiple sclerosis. The diagnosis was provided by the hospital, and the examination was performed by a neurologist. The diagnosis was based on the 2017 McDonald criteria [12]. The study group included patients with: relapsing-remitting multiple sclerosis (RRMS); primary progressive multiple sclerosis (PPMS); secondary progressive multiple sclerosis (SPMS). All patients from the study group underwent a neurological examination and assessment on the Expanded Disability Status Scale (EDSS) [13]. This scale assesses: functions of the pyramidal system, functions of the cerebellum, functions of the brainstem, functions of the sensory system, functions of the intestines and urinary system, visual functions, mental functions, mobility abilities and other irregularities in functioning noted by a doctor. In the next part of the study, the responses of people participating in the study were analyzed in order to select people with current cervical spine pain. All people who marked 'YES' in the original questionnaire were subjected to further examinations of the cervical spine. People who marked ‘NO’ were not examined for cervical spine pain (only the range of motion of the cervical spine was measured) and only completed a quality of life survey. The individual scales and stages of the study are discussed below.
The first part of the cervical spine examination was based on a subjective assessment of pain. The Visual Analogue Scale (VAS) was used to subjectively assess the pain intensity. The interpretation is as follows: a value of 1-3 points for mild pain, 4-7 points for moderate pain, 8-9 points for severe pain, and 10 points meant unbearable pain [14].
In the second part, patients assessed cervical spine pain according to the Laitinen Scale. This scale focuses on four aspects of pain rated from 0 points (no problem) to 4 points (maximum problem). Questions asked to the patient concern: pain intensity, frequency of pain, frequency of taking painkillers and limitation of motor activity. The results obtained are interpreted as follows: 0-4 points (0-25%) - no pain or minimal discomfort, 5-8 points (26-50%) - slight pain, which may affect the use of painkillers and limit activity motor skills, 9-12 points (51-75%) - moderate pain, which significantly affects the taking of painkillers and limitation of physical activity, 13-16 points (76-100%) - severe or unbearable, which occurs very often and causes constant use of painkillers, making normal work and functioning in life impossible [15].
The last stage of the cervical spine examination was the assessment of pain using the Neck Disability Index (NDI). The main purpose of this tool is to assess your current cervical spine pain. It consists of 10 questions about everyday activities, scored from 0 points (no pain while per-forming the task) to 5 points (the worst possible pain you can imagine while performing the task). The scale includes questions about: pain intensity, independence, lifting objects, activities of daily living, reading, headache, concentration, work, driving, sleeping and recreation. The obtained results can be interpreted as follows: 0-4 points (0-8%) - no disability, 5-14 points (10-28%) - mild disability, 15-24 points (30-48%) - disability to a moderate degree, 25-34 points (50-64%) - significant degree of disability, 35-50 points (70-100%) - total degree of disability [16].
The study used devices and methods with proven scientific significance in detecting problems in the cervical spine. To properly interpret the obtained test results, three different scales with different sensitivity and specificity were used. This allows you to reliably look at the patient's problem and check what factors influence the occurrence of pain.
The last part of the study was conducted based on the EQ-5D-5L questionnaire, which takes into account five dimensions of quality of life. The following are assessed: ability to move, self-service, perform everyday activities, feel pain/discomfort; feeling anxious/depressed. The scale ranges from 1 point (no problem) to 5 points (impossibility to perform the activity). The second part of the assessment was based on a Visual Scale Analogue (EQ-VAS, EuroQol Visual Analogue Scale), with which patients self-reported health condition. This scale shows how good or bad a day respondents are having at the time of the survey. The scale ranges from 0 to 100, where 0 - is the worst imaginable health condition and 100 - the best imaginable state of health [17,18].
Point 2: To elaborate further, if your control group consisted of healthy individuals with no MS and no back pain, what is the novelty of your study? The fact that MS is associated with back pain and that both conditions, together or separately, affect quality of life is not novel and is widely known and recognized. On the other hand, if your healthy individuals were people without MS but with back pain, why were there so few of them? If this is the case, all comparisons between the study and control groups are meaningless, as the composition of the control group was inadequate. Please consider addressing this issue.
Response 2: We agree with your comment that our description may have been unclear to you and may have been incorrectly interpreted. Therefore, appropriate changes have been introduced in the Summary and in Chapter 2.3 (Methods) - located above. We would like to clarify and dispel any confusion that you have noticed. 80 people were recruited for the study, of which 40 people constituted the control group and 40 people constituted the study group. Our study used random sampling, so we did not know how many people would report cervical spine pain. After screening, 27 people from the study group (people with MS, aged 20-50 years) reported neck pain and 16 people (without CNS diseases, aged 20-50 years) from the control group. In the case of such a research model, we cannot artificially induce the number of people in the control group to be the same. Moreover, in the scientific literature, researchers often study groups that are not equally numerous. We do not agree with the statement that our comparisons are senseless. The problem of back pain is widely known in the scientific literature, but is it applicable to every section and every disease? Our research study focuses ONLY on the cervical spine. The cervical spine pain is rarely diagnosed in people with MS because most research articles focus on low back pain or ‘spine pain’ in general. Comparison of the group of people with MS with people from the control group shows that other external or environmental factors may influence the occurrence of pain. Therefore, we considered the use of various scales that present the level of cervical spine pain to varying degrees (they have different sensitivity and specificity). If you think that our answer is insufficient, please specify what exactly you expect. Should we equalize the size of the groups, should we change the distribution of data, or should we describe the results of scales, e.g. NDI, in detail for a clearer understanding of the article? Unfortunately, in the current form of the comment, it is difficult for us to comment on it. We don't understand what exactly the change is? Please explain in more detail or let us know if our answer is already sufficient? Please read the new version of the article as there have been numerous changes (including statistical ones).
Point 3: You state that your continuous variables were normally distributed, and therefore you selected parametric tests, which resulted in significant/highly significant differences between your study groups. However, this seems like an attempt to manipulate the data. For most continuous variables, the reported standard deviation is too wide relative to the mean, which suggests a platykurtic data distribution, which by definition is non-normal. In general, with such small comparison groups and an unhealthy population, it is highly unlikely to observe normal data distribution. Even if the data distribution was normal in the control group, it was obviously non-normal in the study group, and to apply parametric tests, both groups should be normally distributed. Please reconsider this approach.
Response 3: We definitely agree with your comment. The description of the statistical tests we presented is too narrow to properly interpret the study results. We have made every effort to fill in the gaps and have included them in section 2.4. (Statistical analysis). Additionally, we re-submitted the data for statistical analysis to make sure everything was counted correctly. We have made changes to the structure of the tables and added additional data (at the request of Reviewer No. 1). We encourage you to familiarize yourself with the changes introduced.
,, Statistical analyzes were performed using Statistica™ version 13.3 (TIBCO Software Inc., Palo Alto, California, USA) and advanced techniques in Microsoft Excel. G*power software version 3.1.9.7 (Düsseldorf, Germany) was used to calculate the statistical power of the sample size, and the total number of participants was calculated to be 72 (effect size = 0.9, error probability α = 0.05, power = 0.9). A total of 88 people took part in the study, of which 8 results were rejected due to formal deficiencies (lack of consent, lack of signature, incomplete documentation, incorrectly completed questionnaire). Of the 80 people participating in the study, 40 people (50.0%) constituted the control group (healthy people without the Central Nervous System diseases), while 40 people (50.0%) constituted the study group (suffering from multiple sclerosis). Statistical description techniques and the Shapiro-Wilk normality test were used to describe groups and variables. The parametric Student's t test was used for statistical analyses, when the variables were normally distributed. If the variables had a non-normal distribution, the following statistical tests were used: U Mann-Whitney test, Wilcoxon test or Kruskall-Wallis test. In order to verify the hypotheses, quantitative variables (e.g. age) were used to compare groups, and to compare groups in terms of nominal and ordinal variables, a contingency table analysis with chi-square analysis was per-formed. The study assessed the magnitude and clinical significance of the results based on effect size statistics. According to Cohen's indices, a value from 0 to 0.20 indicates a negligible effect size, a value from 0.21 to 0.5 indicates a small effect size, a value from 0.51 to 0.80 indicates a medium effect size, and a value > 0.80 indicates a large effect size. P values ≤ 0.05 were considered statistically significant”
Point 4: Subsection 2.3, titled "Methods," needs to be divided into several parts as it is currently messy and difficult to follow. Additionally, the authors need to explain why they used a variety of scales that appear to duplicate each other. What was the rationale for this?
Response 4: Thank you for your valuable attention. We have made changes to Chapter 2.3 (Methods). We also explained the use of different scales during the study, which is included in the body of the article and the answer to point 1.
Round 2
Reviewer 2 Report
Comments and Suggestions for Authors
One of the major issues I raised in the previous round of revisions is the reporting of continuous variables as means and standard deviations, as well as the justification for the use of parametric tests based on this. These parametric tests resulted in significant findings that have affected the interpretation of the results and the conclusions drawn.
In reality, there are many indications that the continuous variables were non-normally distributed, with standard deviations exceeding the mean for more than 20% of the cases. For example, in Table 3, the 'flexion' category shows 12.7 * 100% / 51.3 = 24.75%. Another example can be found in Table 5, under the 'VAS scale' category (4.5 * 100% / 6.3 = 71.42%). I would like to remind the authors that the normality of data distribution needs to be tested both statistically and graphically. In order to utilize parametric tests (which are more prone to identifying significance where it does not exist), both comparison groups must show normal data distribution; furthermore, the authors need to check for homoscedasticity as well.
